# Soil–Plant Interaction Mediated by Indigenous AMF in Grafted and Own-Rooted Grapevines under Field Conditions

Rita Biasi [1], Elena Brunori [1,*], Silvia Vanino [2,*], Alessandra Bernardini [1], Alessia Catalani [1], Roberta Farina [2], Antonio Bruno [2] and Gabriele Chilosi [1]

[1] Department for Innovation Biological, Agrifood and Forest Systems (DIBAF), University of Tuscia, 01100 Viterbo, Italy

[2] CREA Research Centre for Agriculture and Environment, 00184 Rome, Italy

* Correspondence: brunori@unitus.it (E.B.); silvia.vanino@crea.gov.it (S.V.)

**Abstract:** Plant–soil biota represent a unique living system crucial for improving crops' adaptation to climate change. In vineyards, plant–soil relations are mediated by rootstock–scion interaction, with grafted vines being the main plant material employed in vineyard planting. The interaction between two deeply different biological systems such as Vitis vinifera sativa and the American Vitis species may modify vines' adaptation to abiotic stress. The aim of this study was to investigate the physiological response (chlorophyll content—CHL; stomatal conductance—*gs*) of grafted and ungrafted vines and assess the involvement of Arbuscular Mycorrhizal Fungi (AMF) in this response. In two vineyards located in Central Italy, the presence of AMF was assessed in the roots of grafted and ungrafted *cv* 'Aleatico' vines. The morphological traits of AMF and root mycorrhizal colonization differed in the grafted and ungrafted vines. Under limited climate conditions, ungrafted vines showed better leaf resilience traits (high CHL and *gs* values) and larger AMF storage organs (vesicles). On the other hand, the grafted ones—which are more sensitive to climate conditions (lower *gs* and CHL)—involved AMF colonization strategies (greater abundance of arbuscoles and mycorrhizal colonization potential) linked to the improved uptake and transport of water from the bulk soil to the vine. Taken together, these findings highlight different mycorrhizal colonization strategies and storage behaviors in grafted and ungrafted vineyards and with respect to different physical and chemical soil traits.

**Keywords:** abiotic stress; conservative agriculture; gas exchange; rootstock–scion interaction; soil biota

## 1. Introduction

Healthy soil is crucial for ensuring resilient viticulture in the face of biotic and abiotic stressors [1,2]. Climate change impacts the soil microbiome directly [3,4] and indirectly by influencing plants' physiological and morphological responses and root exudate signals [3,5]. In fact, in the rhizosphere, exudates attract soil microbes capable of colonizing a plant's roots and [6] congregate to increase their resistance and resilience to abiotic and biotic pressures. In particular, rhizosphere microbial diversity is a function of the type of plant species, soil properties [7], environmental conditions, and management practices [5,8]. Soil microbial biodiversity represents an important component of wine *terroir*, which is potentially capable of influencing not only berries' composition but also the vine's adaptation to climate change and the prevention of pests and diseases [9,10]. Furthermore, the high-heterogeneity soil attributes within vineyard agroecosystems exhibit distinct microorganism communities that decisively influence microbial recruitment by the root system [11]. Cultivated grapevines are typically grafted on *Vitis* rootstocks of an American origin, such as *Vitis rupestris*, *V. berlandieri*, and *V. riparia*, which are tolerant to phylloxera aphids and are selected according to the pedoclimatic conditions. In sandy soils or soil characterized by high drainage and "lightness" such as volcanic soils, it is

possible to use ungrafted *V. vinifera* varieties because phylloxera is unable to develop or affects the plant in a less virulent way [12]. Rootstocks have been developed for improving plant resistance to adverse climatic or soil conditions and coping with insufficient levels of mineral nutrition [13–15]. Grafting also influences scion vigor; grape yield and quality, mainly via secondary metabolism in the berries; and vine phenology [15–18]. The rootstock influences the distribution and composition of soil microbial communities, including arbuscular mycorrhizal fungi [9,19–24]. Differences in the AMF root community have not only been associated with different species of rootstocks but also with the genotypes of ungrafted plants [25,26].

Due to the lack of information on how soil characteristics influence roots' mycorrhization rate and vines' physiological responses in grafted and ungrafted grapevine roots in the Mediterranean environment, the present research was undertaken, aiming to [1] assess the endemic presence of AMF in vineyards under conservative management practices (organic farming) and the interactions between AMF and soil physical, chemical, and biochemical fertility; [2] evaluate AMF colonization in grapevine (*V. vinifera* L. cv 'Aleatico') roots in the absence (own-rooted vines) or presence of rootstocks (*V. berlandieri* x *V. riparia*); and [3] study the effectiveness of AMF symbiosis to improve drought tolerance in grafted and own-rooted vines under field conditions (Central Italy).

## 2. Materials and Methods

### 2.1. Study Site

The study was conducted in 2021 and 2022 in two vineyards practicing organic farming ("Azienda Agricola Le Coste", Gradoli municipality, Northern Latium, Central Italy) located in the Vulsini Volcanic District (42.645243, 11.864422) (Figure 1a).

This territory is a historical and classic-grape-wine-growing area where the autochthonous cv 'Aleatico' [27] possesses a Protected Designation of Origin (PDO "Aleatico di Gradoli"). Here, the vineyard agrosystems preserve the highly complex ecosystem and habitat structures along their perimeter, thus assuring the maintenance of the landscape's heterogeneity and connectivity and thereby enhancing the resilience of the innate environmental characteristics [28,29].

The two analyzed vineyards were planted in 2004 with the autochthonous landrace 'Aleatico' (AL) trained in the traditional "alberello" vine-training system at a distance of 0.5 m × 1.5 m off the ground (Figure 1b,c). The two analyzed vineyards differed, with one vineyard containing plants for which cv 'Aleatico' was grafted on *V. Berlandieri* x *V. Riparia* rootstock (AL-420A) and the other containing ungrafted plants (own-root vines, AL-ORV). The soil management practices employed were those standard in organic viticulture, with the difference lying in the control of the grasses based on the use of a roller crimper that pushes down the cover crop in the intra-rows and "crimps" the stems to kill the crops, without removing fresh biomass. According to a previous zonation study [30], the tested vineyards are located in the land unit 'Medium and low warmly exposed sides' on soil with a preeminent lithology, i.e., ignimbrite, which is a poorly consistent volcanic material with different textures (ashes and lapillus). According to other research [27], soil texture of the tested vineyards was representative of the typical soil type of Vulsini Volcanic District, which is classified as 'sandy-loam' or 'sandy' soil type and locally denominated 'lapillo'. The climate of the PDO "Aleatico di Gradoli" grape-growing area was identified as Mediterranean with reduced annual rainfall, where drought events were concentrated in summer (June to August) and where bioclimatic indices [31] showed a growing trend of extreme thermal events.

### 2.2. Climate Data and Ombrometric Diagram

Meteorological data (daily average, maximum, and minimum temperature and precipitation) for 2021 and 2022 seasons were obtained in continuum using a weather station, namely, Agrometeorological Service Agency of Latium Region [32], located close to the study site (4.5 km in a beeline) ((x) 244,597, (y) 4,730,680—UTM33). Average monthly precipitation and monthly average temperature were used to elaborate ombrothermic dia-

grams [31]. According to other research [31], meteorological data were used to determine extreme events related to thermal and rainfall regimes during the 2021 and 2022 seasons: (i) the consecutive dry days with no precipitation or precipitation below the threshold of 1 mm, (ii) the number of days with maximum temperature exceeding 30 °C and 35 °C, and (iii) climate classification according to Tonietto and Carbonneau (2004) [33].

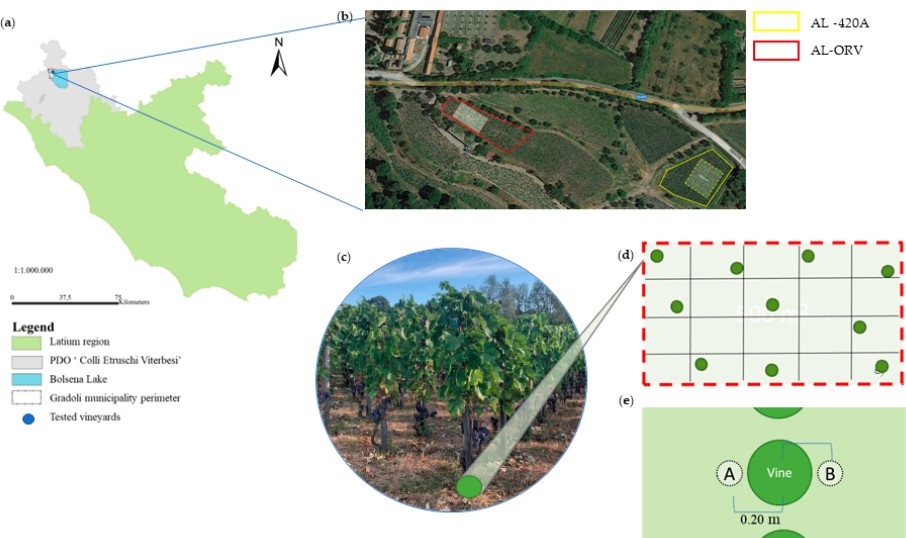

**Figure 1.** (**a**) PDO 'Aleatico di Gradoli' wine-grape-growing area in Central Italy (Latium Region, Gradoli Municipality) and (**b**) localization of the two analyzed vineyards planted with the *cv* 'Aleatico' (AL) variety grafted with *V. berlandieri* x *V. riparia* (rootstock 420A, yellow polygon; AL-420A) and the ungrafted variety (own-root vines, red polygon; AL-ORV) (**c**). Experimental designs adopted for undisturbed soil sample collection around 10 representative vines (**d**) were tested in each vineyard at a distance of 0.20 m from grapevine trunk (two subsamples, A and B, for each vine) and (**e**) at depths of 0–0.20 m.

### 2.3. Samples Collection

In this study, data were collected in each vineyard (AL-ORV and AL-420A) during the 2021/22 growing seasons in autumn (November 2021) when root growth and colonization rates by AMF are still high [29,34]. Ten vines for each vineyard were randomly selected (Figure 1d), and undisturbed soil samples were collected at depths of 0–0.20 m using manual augers (2 subsamples per point at a distance of 0.20 m around grapevine trunk) and introduced in labeled plastic bags for transportation and subsequent processing (drying, sieving, and analysis). From each soil sample, soil and rootlets were separated via sieving; the former was submitted to chemical-physical analyses and the latter to mycorrhization analysis. The extracted roots were carefully cleaned with tap water and stained for mycorrhiza analyses.

### 2.4. Soil Physical and Chemical Parameters

Samples for soil analysis were dried after removing all visible roots and coarse fragments (>2 mm) until reaching constant weight and then sieved with 2 mm sieves and stored at room temperature until analysis. The following determinations were carried out for these samples: soil pH, which was measured in $H_2O$ using a pH meter (inoLab® Multi 9310 IDS); total organic carbon (TOC g·kg$^{-1}$), which was measured using a LECO TOC Analyzer, mod. RC-612 (LECO Corporation, 1987); and total N, which was measured using (N tot g·kg$^{-1}$) LECO Nitrogen analyzer FP-528 (St. Joseph, MI, USA). The C/N ratio index was calculated with reference to the TOC and N content.

The following properties were determined by performing the corresponding analyses on the undisturbed samples: soil texture via particle size analysis (sand, silt, and clay percentage); bulk density via core-sampling method (g·cm$^{-3}$); and soil water content at wilting point (SWW) and soil water content at field capacity (SWFC), which were calculated

using the retention curve method [35], in which moist samples were dried by raising the air pressure in an extractor with a porous ceramic plate.

### 2.5. Estimation of Mycorrhizal Root Colonization

The level of mycorrhizal colonization (MyCP), the frequency of mycorrhizal colonization (F-%), the intensity of mycorrhizal colonization (M-%), and the abundance of arbuscules (A-%) and vesicles (V-%) were estimated using the method described by Trouvelot [36]. The root system, now free of soil, was washed several times with water, and 1 cm long roots were soaked with KOH (10%) for 5 min at 92 °C. Roots were washed with distilled water, stained with 5% ink–vinegar solution (to stain all fungal structures) for 5 min at 92 °C, and then destined by rinsing in tap water [37]. Root fragments were mounted on slides with a glycerol drop and observed under a microscope (Axioskop Zeiss, Germany); then, fungal hyphens, arbuscules, and vesicles were photographed. First, degrees of colonization and abundance of arbuscules and vesicles were calculated according to the range of Trouvelot's classes; afterward, the results relating to the abundance of arbuscules and mycorrhizal colonization were used to estimate mycorrhizal frequency and mycorrhizal intensity parameters according to Trouvelot's formulae.

### 2.6. Physiological Parameters—Stomatal Conductance and Chlorophyll Content

Physiological traits of leaves of own-rooted 'Aleatico' vines (ALV-ORV) and grafted (AL-420A) were monitored during the 2021 and 2022 seasons. Chlorophyll content (Chl—$\mu$mol of chlorophyll per $m^2$ of leaf surface) and stomatal conductance ($gs$—mmol $H_2O$ $m^2$ $s^{-1}$) of leaves were assessed for the ten randomly selected vines of each vineyard using user-friendly and field-portable devices, namely, a MC-100 Chlorophyll Concentration Meter (Apogee Instruments, Inc., Logan, UT, USA) and a Leaf porometer (SC-1, Decagoon Devices, Pullman, WA, USA), respectively. Three measurements for each vine were performed on five basal mature and fully exposed leaves; for each leaf, three lectures were conducted: two close to the base of the petiole sinus and one on the tip. Parameter monitoring was performed from the flowering phenological phase (BBCH 065) to the ripening of berries (BBCH 089).

### 2.7. Statistical Analysis

Raw data ($n = 10$ per each tested parameter) with a normal distribution (Kolmogorov–Smirnov test; $p > 0.05$) and homogeneity of variance (Levene's test; $p > 0.05$) were analyzed using one-way analysis of variance. The multiple comparison Newman–Keuls test was used to assign significant differences between means ($p$-value $\leq 0.05$). The Pearson correlation test was used to determine the relationship between all measured soil and mycorrhizal parameters.

In order to identify multicollinearity in set data, High Variance Inflation Factor (VIF) and Low Tolerance method were employed. A new set of variables, which were independent of each other, were selected according to VIF. This set was analyzed via principal component analysis (PCA). PCA was carried out with the aim of investigating the relationship between AMF functional traits, soil chemical and physical traits, and genotypes (*V. berlandieri* x *V. riparia*, AL-420A and *V. vinifera* L. AL-ORV). Relevant components with eigenvalue > 1 were chosen after computation of the correlation matrix.

The Keiser–Meyer–Olkin (KMO) measure of sampling adequacy was employed, which tests whether the partial correlations between variables are small. A scatterplot was developed to compare the distribution of component scores with factor loadings. All data were analyzed using XLSTAT software v.2022 2.1 (Addinsoft) [38].

## 3. Results

### 3.1. Climate Analysis

During the two observed growing seasons (2021 and 2022), abiotic stressors in the grape-growing area of 'Aleatico' PDO were evaluated. Bagnouls–Gaussen climatic dia-

grams (Figure S1) and extreme climate events were analyzed. There were 210 and 187 dry days (absence of precipitation) during the 2021 and 2022 growing seasons (from 1 April to 31 October), respectively, with total amounts of rainfall equal to 376.20 mm and 367.80 mm, respectively. The dry day distribution was as follows: there were 24 dry days during the bud development period (March–April), 80 dry days from inflorescence emergence (May) to berry set (July), and 83 dry days from the ripening of the berries (August) to leaf-fall (October) for the 2021 growing season, while there were 29 dry days during the bud development period (March–April), 83 dry days from inflorescence emergence (May) to berry set (July), and 23 days during the berry-ripening period (August) for the 2022 growing season. Regarding extreme thermal events, there were 80 and 90 days in which the maximum temperature exceeded 30 °C and 21 and 36 days in which the maximum temperature exceeded 35 °C in the 2021 and 2022 seasons, respectively.

### 3.2. Soil Analysis

The descriptive traits for the physical and chemical soil indicators of the two analyzed vineyards, referring to the grafted (AL-420A) or own-root vines of cv 'Aleatico' (AL-ORV), are reported in Table 1. The two analyzed vineyards significantly differed in terms of sand, loam, and clay content; C/N ratio; and soil water content at field capacity (SWFC %) in the topsoil (0–0.20 m depth), while regarding the soil pH, TOC, N, bulk density, wilting point, and porosity, no significant differences were shown. Nonetheless, according to the proportions of sand, silt, and clay-sized particles in the soil, the texture for both vineyards could be classified as sandy loam. TOC content was above the threshold of 2% for both vineyards. C/N showed the highest value in the AL-420A vineyard, which was almost twice the value measured for the AL-ORV one.

**Table 1.** Physical and chemical traits of soil for the two analyzed vineyards, AL-420A and AL-ORV, with grafted and own-root vines of the cv 'Aleatico', respectively. Asterisks represent statistical significance for comparison between vineyards (** $p \leq 0.01$, and * $p \leq 0.05$; *ns*—not significant). (TOC: Total Organic Carbon; TN, Total Nitrogen; C/N: Carbon/Nitrogen ratio; FWC: Field Water Capacity; WP: Wilting Point).

| Soil Chemical and Physical Parameters | AL-ORV | AL-420A | *Signif.* |
|---|---|---|---|
| Sand (%) | 53.35 | 48.55 | * |
| Silt (%) | 34.55 | 44.05 | ** |
| Clay (%) | 12.1 | 7.4 | ** |
| TOC (%) | 2.49 | 2.63 | *ns* |
| N (%) | 0.5 | 0.34 | *ns* |
| C/N ratio | 4.7 | 7.27 | ** |
| Bulk density (BD) | 0.93 | 0.93 | *ns* |
| Soil water content at field capacity (SWFC %) | 22.46 | 27.51 | ** |
| Soil water content at wilting point (SWW %) | 15.15 | 14.86 | *ns* |
| Porosity (P %) | 65.01 | 64.65 | *ns* |
| pH | 6.8 | 6.9 | *ns* |

### 3.3. Mycorrhizal Colonization

A microscopic investigation showed that AMFs were always present in the topsoil of the tested vineyards. In the vine roots, AMFs presented typical mycorrhizal structures (Figure 2). The mean values of mycorrhizal colonization (MyCP-%), mycorrhizal frequency (F-%), intensity of mycorrhizal root colonization (M%), and relative arbuscular richness (A-%) and vesicles (V-%) determined for each root sample are reported in Table 2.

Roots from the grafted vines (AL-420A), i.e., roots of *V. berlandieri* x *V. riparia* rootstock, exhibited statistically higher values of mycorrhizal colonization (MyCP-%), mycorrhizal frequency (F-%), intensity of mycorrhizal root colonization (M-%), and relative arbuscular richness (A-%), whereas the roots of *Vitis vinifera* L. (AL-ORV) showed a significantly higher percentage of richness in vesicles (V-%).

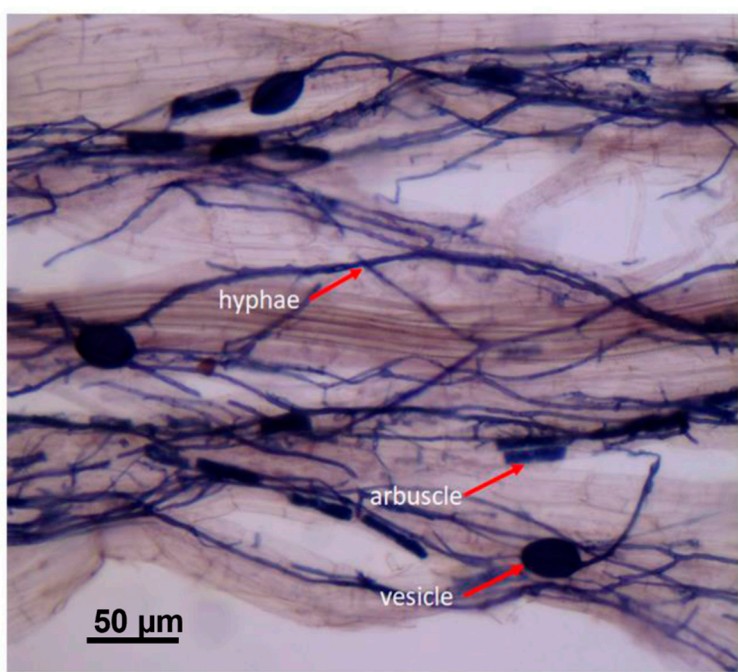

**Figure 2.** Typical AFM mycorrhizal structures from sampled grape roots stained with ink–vinegar solution from the scion–rootstock combination (AL-420A) or the own-rooted vines (AL-ORV). Bar = 50 μm.

**Table 2.** Mycorrhizal colonization (MyCP-%), frequency of mycorrhizal colonization (F-%), intensity of mycorrhizal colonization (M-%), and frequency of arbuscules (A-%) and vesicles (V-%) in mycorrhizal root fragments sampled in the vineyard with grafted vines of *cv* 'Aleatico' (AL-420A) and in vineyard with own-rooted vines (AL-ORV). Asterisks represent statistical significance for comparison between vineyards (a, b $p \leq 0.05$; *ns*—not significant).

| Vineyards | MyCP | F | M | A | V |
|---|---|---|---|---|---|
| AL-ORV | 77.3 b | 90.0 *ns* | 73.4 b | 44.5 b | 16.9 a |
| AL-420A | 86.7 a | 93.3 *ns* | 82.8 a | 54.0 a | 12.7 b |

### 3.4. Vine Physiological Responses

The physiological responses of the two types of vines, i.e., grafted and own-rooted, to the increasing levels of drought and the hot climate during the growing season were evaluated by measuring the leaf chlorophyll content (CHL) and stomatal conductance (*gs*) as indicators of the soil–plant interactions mediated by native AMF (Table 3).

CHL showed significant differences during all phenological stages in the 2022 season and until fruit set during the 2021 season (BBCH 079). In both seasons, the CHL for the AL-ORV vines was higher than that of the AL-420A vines.

Stomatal conductance showed very low values from BBCH 079 to BBCH 089 for both seasons due to limited water conditions characterized by long to very long dry spell durations (Figure S1 in the Supplementary Materials). These conditions continued until the leaves had fallen in 2021 and, combined with extreme temperature events, led to stomatal closure.

### 3.5. Statistical Analyses

The PCA extracted two components with more than 69% cumulated variance for all the investigated physical, chemical, and biological (mycorrhizal functional traits) parameters. The distribution of component loadings and scores is shown in Figure 3. Component 1 (F1) represents a gradient of plant genotype; in particular, grafted vines (AL-420A) are in quadrants I and II and own-rooted vines of *cv* 'Aleatico' are in quadrants III and IV, indicating genotype-base effects. Concerning soil physical traits, the clay content (%) in

the topsoil showed a significant negative Pearson correlation coefficient with mycorrhizal colonization parameters (Table 4), such as intensity of mycorrhizal colonization (M) ($-0.57$) and frequency of arbuscules (A) ($-0.66$), and with soil water content at field capacity (SFWC) ($-0.62$), C/N ratio (-0.63), and TOC ($-0.45$). Positive Pearson correlation coefficients were shown for clay content and vesicles (V) (0.69) and between C/N ratio and soil water content at field capacity (SFWC) (0.58), intensity of mycorrhizal colonization (M) (0.55), and abundance of arbuscules (A) (0.61). Positive Pearson correlation coefficients were also shown between (i) soil water content at field capacity (SFWC) and frequency of arbuscules (A) (0.45) and (ii) pH and MyCP (0.47), M (0.55), and A (0.55).

**Table 3.** Chlorophyll content (CHL—µmol of chlorophyll per $m^2$ of leaf surface) and stomatal conductance ($gs$—mmol $H_2O$ $m^2$ $s^{-1}$) of leaves from vines of the two analyzed vineyards (AL—420A and AL-ORV vineyards with grafted and own-rooted vines of $cv$ 'Aleatico', respectively) according to grapevine phenological stages (BBCH 065—full flowering; BBCH 071—fruit set: fruits begin to swell and remains of flowers lost; BBCH 079—majority of berries touching; BBCH 089—berries ripe for harvest). Asterisks represent statistical significance for comparison between vineyards (** $p \leq 0.01$, and * $p \leq 0.05$; $ns$—not significant).

| | | CHL | | | $gs$ | | |
|---|---|---|---|---|---|---|---|
| | **Phenological Stages** | **AL-ORV** | **AL_420A** | *Sign.* | **AL-ORV** | **AL_420A** | *Sign.* |
| | BBCH065 | 196.1 ± 22.3 | 173.5 ± 16.5 | ** | 214.9 ± 27.5 | 186.8 ± 24.2 | ** |
| 2021 | BBCH071 | 278.8 ± 37.0 | 230.7 ± 34.7 | ** | 425.9 ± 41.7 | 384.5 ± 56.5 | ** |
| | BBCH079 | 288.7 ± 36.7 | 274.1 ± 54.0 | $ns$ | 220.6 ± 46.2 | 186.6 ± 39.3 | $ns$ |
| | BBCH089 | 298.5 ± 36.4 | 317.5 ± 43.2 | $ns$ | 178.3 ± 90.7 | 78.2 ± 62.9 | ** |
| | BBCH065 | 305.6 ± 14.1 | 241.5 ± 16.1 | ** | 397.4 ± 65.5 | 365.5 ± 98.0 | $ns$ |
| 2022 | BBCH071 | 395.2 ± 19.6 | 319.2 ± 16.1 | ** | 338.9 ± 12.1 | 321.1 ± 18.9 | $ns$ |
| | BBCH079 | 376.7 ± 16.1 | 32406 ± 11.2 | ** | 219.1 ± 60.4 | 208.3 ± 83.6 | $ns$ |
| | BBCH089 | 400.6 ± 17.0 | 346.8 ± 11.6 | ** | 201.1 ± 10.7 | 189.7 ± 10.9 | * |

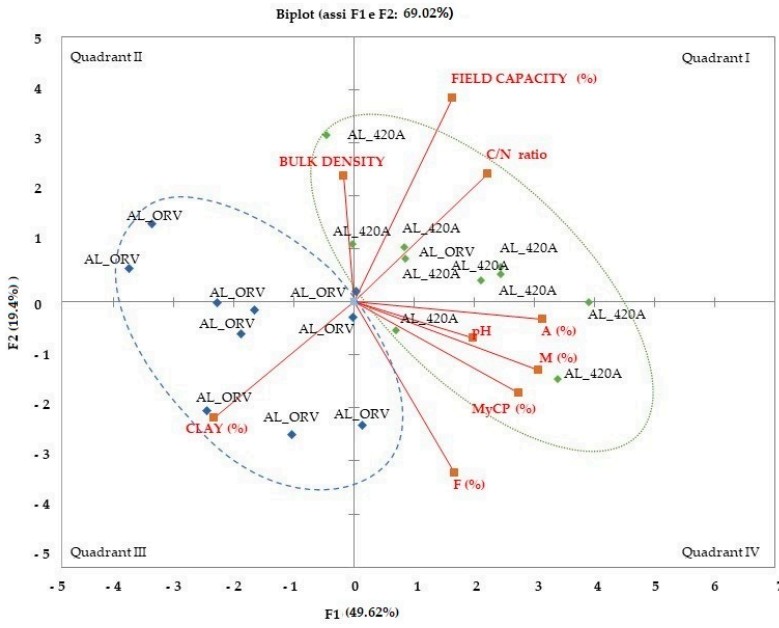

**Figure 3.** Factor loadings and scores of PCA regarding mycorrhizal colonization (MyCP-%); frequency of mycorrhizal colonization (F-%); intensity of mycorrhizal colonization (M-%); frequency of arbuscules (A-%) and vesicles (V-%) in mycorrhizal root fragments sampled from grafted vines of cv 'Aleatico' vines (AL-420A) and in vineyard with own-rooted $cv$ 'Aleatico' vines (AL-ORV); and soil physical and chemical traits for the two analyzed vineyards (C/N—carbon/nitrogen ratio; clay content; field capacity; bulk density; pH).

**Table 4.** Correlation matrix (Pearson coefficient) concerning soil physical and chemical traits for the two analyzed vineyards (TOC: Total Organic Carbon; N total nitrogen; C/N: carbon/nitrogen ratio) and mycorrhizal colonization (MyCP-%), frequency of mycorrhizal colonization (F-%), intensity of mycorrhizal colonization (M-%), and frequency of arbuscules (A-%) and vesicles (V-%) in mycorrhizal root fragments sampled in vineyard with grafted vines of *cv* 'Aleatico' (AL-420A) and in vineyard with own-root vines (AL- ORV). Bold values are different from 0 at the alpha = 0.05 significance level.

| Variables | Sand (%) | Loam (%) | Clay (%) | TOC (%) | N (%) | C/N Ratio | Bulk Density | Soil Water Content at Field Capacity (SWFC %) | Soil Water Content at Wilting Point (SWW %) | Porosity (%) | pH | MyCP (%) | F (%) | M (%) | A (%) | V (%) |
|---|---|---|---|---|---|---|---|---|---|---|---|---|---|---|---|---|
| Sand (%) | 1 | | | | | | | | | | | | | | | |
| Loam (%) | **−0.92** | 1 | | | | | | | | | | | | | | |
| Clay (%) | 0.30 | **−0.66** | 1 | | | | | | | | | | | | | |
| TOC (%) | −0.04 | 0.22 | **−0.45** | 1 | | | | | | | | | | | | |
| N (%) | 0.017 | 0.04 | −0.13 | −0.30 | 1 | | | | | | | | | | | |
| C/N ratio | −0.36 | **0.55** | **−0.63** | **0.70** | **−0.52** | 1 | | | | | | | | | | |
| Bulk density (BD) | −0.20 | 0.14 | 0.04 | −0.09 | 0.02 | −0,07 | 1 | | | | | | | | | |
| Soil water content at field capacity (SWFC %) | **−0.62** | **0.74** | **−0.62** | 0.24 | 0.04 | **0.58** | 0.40 | 1 | | | | | | | | |
| Soil water content at wilting point (SWW %) | −0.17 | 0.07 | 0.15 | −0.20 | 0.18 | −0.28 | 0.42 | 0.20 | 1 | | | | | | | |
| Porosity (P %) | 0.17 | −0.11 | −0.05 | 0.09 | −0.13 | 0.13 | **−0.54** | −0.25 | **−0.75** | 1 | | | | | | |
| pH | −0.15 | 0.25 | −0.31 | **0.56** | 0.02 | 0.41 | −0.22 | 0.14 | −0.08 | −0.01 | 1 | | | | | |
| MyCP (%) | 0.14 | 0.05 | −0.38 | 0.19 | 0.08 | 0.34 | 0.04 | 0.21 | −0.36 | 0.05 | **0.47** | 1 | | | | |
| F (%) | 0.06 | 0.00 | −0.10 | −0.24 | 0.15 | −0.05 | −0.08 | −0.03 | −0.35 | −0.01 | 0.16 | **0.65** | 1 | | | |
| M(%) | 0.02 | 0.22 | **−0.57** | 0.42 | −0.03 | **0.55** | −0.11 | 0.27 | −0.43 | 0.14 | **0.55** | **0.92** | 0.63 | 1 | | |
| A(%) | −0.16 | 0.40 | **−0.66** | 0.37 | 0.04 | **0.61** | −0.06 | **0.45** | −0.37 | 0.09 | **0.55** | **0.86** | **0.53** | **0.93** | 1 | |
| V(%) | 0.39 | **−0.59** | **0.69** | −0.08 | −0.27 | −0.372 | 0.08 | **−0.58** | −0.157 | 0.18 | −0.05 | −0.05 | 0.13 | −0.09 | −0.21 | 1 |

## 4. Discussion

Soil microbial diversity regulates the performance and interactions between plants and the rhizosphere [39,40]. In viticulture, the soil microbiome, comprising fungi and bacteria, contributes to grapevines' adaptation to climate change, improving vine resilience and limiting environmental stresses [2]. In particular, arbuscular mycorrhizal fungi, also referred to as biofertilizers, have been identified as being capable of promoting the tolerance and resistance of host plants to abiotic stress factors such as drought and extreme temperatures [41].

Understanding the interactions between plants and rhizosphere microbes under real field conditions constitutes a tipping point towards the maintenance of the community structures and diversity of native arbuscular mycorrhizal fungi, thereby favoring plant stress tolerance [42,43], the phytoremediation of contaminated soils [44,45], crop health, and productivity [3].

The findings of this field-based observational study showed that AMF relative abundance (MyCP-%) and AMF structure varied not only according to soil chemical and physical traits but also as a function of the type of *Vitis* species, i.e., *V. vinifera* vs. *V. berlandieri* x *riparia*, used as a rootstock. The type of soil has been recognized as a key determinant of the species composition and richness of AMF communities [46,47]. According to other researchers [48], a positive correlation between soil pH, mycorrhizal colonization (MyCP-%), intensity of mycorrhizal colonization (M-%), and frequency of arbuscules (A-%) has been found. Soil texture, hydrological soil indices, and, particularly, clay content and field water capacity (FWC) were confirmed as determinants of the abundance and richness of AMF species [49]. In both tested vineyards, soils with coarse (sandy) particles and that were loose and well-drained were associated with an increase in mycelial growth, stimulating mycorrhizal colonization due to their greater porosity and lower FWC. In fact, under low FWC (<40%) [50], and thus under limiting water conditions, AMF support plant resilience by favoring root growth [51,52], developing extensive root systems [53], and mediating drought stress tolerance in plants. One possible mechanism behind this process relates to the ability of AMF hyphae to increase water and nutrient uptake by plants as a result of the symbiotic relationship between plants and AMF [6,53].

In semi-arid conditions, such as those in Mediterranean environments, AMF diversity preservation could be considered as a method for driving plants toward adaptation to climate change. At the same time, the preservation of this agrobiodiversity may offer multiple ecosystem services, such as reducing water consumption in agrosystems, improving water use efficiency [54], and increasing leaves' photochemical efficiency [55], i.e., a crop's carbon storage capability. Mycorrhizal vines, especially under limiting abiotic conditions, present beneficial effects with respect to the assimilation of $CO_2$, improving and regulating the use of water, and increasing photosynthetic rates [56]. In viticultural systems, all these benefits are mediated by rootstocks that differ in terms of genotype and root system conformation, modulating the response of grapevine to abiotic and biotic conditions.

The present study pointed out the structural and functional differences between mycorrhized 'Aleatico' vines grafted on a phylloxera-tolerant rootstock (420A—*V. berlandieri* x *V. riparia*) or on own-rooted variety (*V. vinifera*) in terms of AMF traits and physiological responses. In particular, the differences observed regarding chlorophyll breakdown and leaf gas exchange (conductance) can be considered vine resilience traits [2].

The simultaneous presence of a fungal network, arbuscules, and vesicles highlights the presence of an equilibrated colonization strategy, with all colonization strategies—proliferative, storage, and transfer—evidenced for both mycorrhizal 'Aleatico' vines. In particular, the differences in mycorrhizal colonization (MyCP-%), frequency and intensity of mycorrhizal colonization (F and M-%), and occurrence of arbuscules (A-%) and vesicles (V-%) in the mycorrhizal root fragments of the grafted vines (AL-420A) compared to the own-rooted vines (AL-ORV) confirmed the role of rootstock genotype as one of the major bio-factors able to influence AM root colonization and the composition and species richness of AMF communities. Grafted plants showed higher values for all the mycorrhizal colonization

parameters related to the greater presence of arbuscules compared to vesicles, which may be associated with faster root turnover and a higher demand for nutrient transfer from AM fungi to plants [57]. The greater presence of vesicles observed in the ungrafted vines can be interpreted as a storage strategy, which consists of a better adaptation of these vines (greater stomatal conductance and higher leaf chlorophyll content) to extreme climate conditions [58]. The observed higher stomatal conductance, however, implies increased water consumption; thus, the presence of the rootstock seems to be more beneficial in terms of sustainable cultivation.

In order to drive the agro-food sector towards providing sustainable food systems, the integration of agroecological techniques for enriching or maintaining telluric microbial biodiversity and, in particular, enhancing indigenous populations of AMF [59] will be crucial because they can be considered to constitute bioprotective biological systems against climate alterations and a strategy for increasing resilience and biodiversity in intensive cropping systems [60,61].

## 5. Conclusions

Ongoing climate change is significantly impacting the grape wine sector. Plant–mycorrhizal symbiosis have great potential in environmentally friendly agriculture, modulating nutrient cycling, alleviating abiotic and biotic stress, mediating the plant–soil relationship, and preserving yield and plant performance.

Therefore, a better understanding of the plant–soil relationships affected by these intermediary biological organisms is the basis for the identification of sustainable agronomical practices.

This research has examined the soil–plant interactions mediated by AMF symbiosis in vineyards, wherein the vines' physiological responses under sensitive environments proved that both the type of vineyard soil and the grapevine root system genotype affect AMF typology and colonization strategies. In terms of water use and carbon storage capacity, as expressed by stomatal conductance and chlorophyll accumulation, the most efficient plant–soil system proved to be the grafted mycorrhizal plant material.

The possibility of inducing the symbiotic relationship with AMF in a nursery by using autochthonous fungi endemic in the soil would allow not only the availability of plant material of higher quality for planting in vineyards but also likely the maintenance of the symbiotic relation over time, thereby opening up a wide range of potential applications in sustainable agrifood systems.

**Supplementary Materials:** The following supporting information can be downloaded at: https://www.mdpi.com/article/10.3390/agriculture13051051/s1. Figure S1: Climatic Bagnouls and Gaussen diagram and dry periods (red area) for the season 2021 (a) and 2022 (b) in tested grape-wine growing area (Central Italy–Gradoli municipality).

**Author Contributions:** Conceptualization, R.B. and E.B.; methodology, R.B., E.B., G.C., S.V. and R.F.; data collection, A.B. (Alessandra Bernardini), A.C. and A.B. (Antonio Bruno); data analysis, E.B. and A.B. (Alessandra Bernardini); writing—original draft preparation, R.B., G.C., E.B. and S.V.; funding acquisition, R.B. and S.V. All authors have read and agreed to the published version of the manuscript.

**Funding:** Research funded by the PRIMA Programme supported by the European Union, GA n°2041—LENSES Project, and by the Regione Lazio–LAZIO INNOVA—under the Research Groups 2020 (co-funding European POR-FESR 2014-2020)—project MICOVIT, Grant number 107948-0300-0327. The ACP was funded by CREA-AA. This paper and the content contained therein do not represent the opinion of the PRIMA Foundation; the PRIMA foundation is not responsible for any uses of its content.

**Institutional Review Board Statement:** Not applicable.

**Data Availability Statement:** Not applicable.

**Acknowledgments:** The authors thank the owners of the "Le Coste" farm (Gradoli, Viterbo Italy) for having hosted the research in its vineyards.

**Conflicts of Interest:** The authors declare no conflict of interest.

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
