# Peer review of "Soil–Plant Interaction Mediated by Indigenous AMF in Grafted and Own-Rooted Grapevines under Field Conditions"

_agriculture, doi:10.3390/agriculture13051051_

Round 1

Reviewer 1 Report

The manuscript entitled "Soil-plant interaction mediated by indigenous AMF in grafted and own-rooted grapevines under field conditions" by Biasi et al. investigate the physiological response of grafted and ungrafted vines by the presence of Arbuscolar Micorrizhal Fungi (AMF). This work is significance and material and methods are well structured and written. In general, the manuscript is well prepared and clearly presented. However, I have some suggestions for the authors in order to improve the manuscript. Please see attachement with comments.

Reviewer 2 Report

Soil-plant interaction mediated by indigenous AMF in grafted and own-rooted grapevines under field conditions.

Title: No full-stop in the title

Abstract section is very introductory. Present some important results and findings in the abstract.

Manuscript write-up makes it confusing about site, field and variety. Rephrase manuscript to make it clear, concise and easily understandable.

L134: “root washed 3-times or roots washed thoroughly”

L 107 /184: Add/remove space according to the journal format ’92oC or 30 oC’

L140: Mention how did you estimate AMF by adding 1 or 2 sentences.

L148: remove extra space after ‘each’. Check extra-space issue throughout the manuscript.

Table 1, 2 and 3: Table footnote is placed with the table title. Correct it.

L227: Figure 2. T should be capital in ‘typical’

Figure 2: add scale with the micrograph. Description is missing in the figure legend.

Table 1 and 2: ISO4 abbreviation of Significance is “Signif” (Oxf).

Discussion: Start with a sentence on “microbial diversity”, then come to AMF diversity.

L297-299: rephrase

L303: add space after comma ‘(M - %),frequency’

Conclusion: Shorten the conclusion section.

Add some latest references 2020-2023 to introduction and discussion

English Quality is above average, but still needs improvement.
